# Adenovirus Receptor Expression in Cancer and Its Multifaceted Role in Oncolytic Adenovirus Therapy

**DOI:** 10.3390/ijms21186828

**Published:** 2020-09-17

**Authors:** Lobke C.M. Hensen, Rob C. Hoeben, Selas T.F. Bots

**Affiliations:** 1Master Research Program Infection and Immunity, Faculty of Medicine, Utrecht University, 3584 CS Utrecht, The Netherlands; l.c.m.hensen@students.uu.nl; 2Virus and Cell Biology Lab, Department of Cell and Chemical Biology, Leiden University Medical Center, 2333 ZC Leiden, The Netherlands; r.c.hoeben@lumc.nl

**Keywords:** oncolytic adenovirus therapy, human adenovirus, receptor expression, CAR, CD46, DSG-2, sialic acid, integrins

## Abstract

Oncolytic adenovirus therapy is believed to be a promising way to treat cancer patients. To be able to target tumor cells with an oncolytic adenovirus, expression of the adenovirus receptor on the tumor cell is essential. Different adenovirus types bind to different receptors on the cell, of which the expression can vary between tumor types. Pre-existing neutralizing immunity to human adenovirus species C type 5 (HAdV-C5) has hampered its therapeutic efficacy in clinical trials, hence several adenoviral vectors from different species are currently being developed as a means to evade pre-existing immunity. Therefore, knowledge on the expression of appropriate adenovirus receptors on tumor cells is important. This could aid in determining which tumor types would benefit most from treatment with a certain oncolytic adenovirus type. This review provides an overview of the known receptors for human adenoviruses and how their expression on tumor cells might be differentially regulated compared to healthy tissue, before and after standardized anticancer treatments. Mechanisms behind the up- or downregulation of adenovirus receptor expression are discussed, which could be used to find new targets for combination therapy to enhance the efficacy of oncolytic adenovirus therapy. Additionally, the utility of the adenovirus receptors in oncolytic virotherapy is examined, including their role in viral spread, which might even surpass their function as primary entry receptors. Finally, future directions are offered regarding the selection of adenovirus types to be used in oncolytic adenovirus therapy in the fight against cancer.

## 1. Introduction

Cancer remains one of the most important causes of death in the Western world. For most tumor types there is still no cure, hence there is a dire need for new and effective ways of treatment. A promising new anticancer therapy is oncolytic virus therapy (OVT). The idea to use viruses to specifically target and kill tumor cells was derived from early observations during the 19th century [1]. Patients with a tumor sometimes went into spontaneous remission upon contraction of an infectious agent. Oncolytic viruses either have a natural preference for tumor cells or are genetically engineered to preferentially infect, replicate in, and kill tumor cells [2]. The virus-induced killing of tumor cells is mediated in two ways. First, viruses induce direct lysis of tumor cells upon viral replication. Second, upon immunogenic cell death of the tumor cell, the immune system is activated. This results in an antitumor immune response, where cytotoxic T cells specifically target and kill tumor cells, including noninfected tumor cells [3].

Adenoviruses (Ads) are one of the most studied oncolytic viruses. Ads are nonenveloped viruses with a large double-stranded DNA genome of roughly 36 kB in size. As of today, 103 human adenovirus (HAdV) types are known based on their genotype (http://hadvwg.gmu.edu/) and more types might be discovered in the future [4]. The 103 HAdVs are divided into seven ‘species’ named A to G, formerly called ‘subgroups’ [5]. Recent classification of Ads also uses their genome sequence for classification [4], hence, the term ‘serotypes’ is replaced by ‘types’. Human Ad types are endemic in the human population and according to the World Health Organization, 2–5% of all the common colds can be attributed to Ad infections [6]. HAdVs from different species are associated with different clinical manifestations such as gastroenteritis, which is mainly caused by HAdV-F and -G species [7,8], or keratoconjunctivitis, which is primarily the result of HAdV-D species infections [9]. This diversity is presumably related to the use of different entry receptors between HAdV species. The seroprevalence of Ad types and species in the population differs, with some types classified as species A and C being the most prevalent [10].

The high seroprevalence of certain HAdV types consequently leads to pre-existing neutralizing immunity against these types. This pre-existing immunity to certain Ad types is believed to have resulted in moderate and variable results in clinical trials when using adenoviral vectors. Pre-existing immunity can be divided into a cellular arm and a humoral arm. Cellular pre-existing immunity entails T cell immunity against antigens that are often shared by multiple Ad types [11]. In contrast, pre-existing neutralizing antibodies (nAbs) are directed against HAdV regions that are more type specific. The nAbs reduce the efficacy of OVT by blocking viral infection and thus Ad delivery to the tumor microenvironment (TME). Therefore, nAbs mainly hinder systemic administration of oncolytic HAdVs as opposed to intratumoral injection of viral vectors [3]. In support of this, a recent study indicated that nAbs were not capable of effectively neutralizing Ad particles to prevent infection when the virus was administered locally [12]. For systemic administration, there is as of yet no consensus regarding the effect of nAbs on Ad therapy due to the widely contradicting results in preclinical and clinical studies [13]. Regardless, systemic administration is still favored over direct injection in the tumor, as not all tumors can be directly injected. In addition, metastasized tumors could possibly be targeted. Systemic administration of oncolytic adenoviral vectors thus broadens the applicability of the therapy. While the effect of nAbs on systemic administration of adenoviral vectors remains to be further elucidated, it could be desirable to evade pre-existing immunity in order to create a more homogenous patient population. This might reduce the observed patient-to-patient variation in clinical trials, regardless of the effect of nAbs.

One way of avoiding neutralizing pre-existing immunity is by using less seroprevalent HAdV types [11]. So far, most research has focused on vectors derived from human Ad species C type 5 (HAdV-C5). The pre-existing immunity against HAdV-C5 in the population may underlie the moderate efficacy of oncolytic Ad therapy. However, the use of another, less seroprevalent HAdV as a vector could overcome the issues observed with pre-existing immunity against HAdV-C5 [10]. Therefore, the use of low seroprevalent oncolytic HAdVs could be used to circumvent pre-existing immunity in OVT as well. Besides less prevalent HAdVs, Ads from another host species, such as non-human primates, can be used to circumvent the pre-existing immunity to HAdVs [14]. Non-human primate Ads especially are considered because of their low pathogenicity in their natural host and their genome similarity to HAdVs [15]. In addition, human cells are permissive to many of the Ad types isolated from the great apes. Non-human primate Ads do not circulate in the human population and therefore nAbs against these viruses are less prevalent, which is a major benefit of using non-human primate adenoviral vectors [15]. However, the receptors used for entry into the human cell by some low prevalent human and non-human primate Ads are not always known and might differ from HAdV-C5.

The development of adenoviral vectors based on other (sero)types than HAdV-C5 raises the question whether these viruses are suitable for targeting cancerous cells. As mentioned earlier, different HAdV species can bind to different cellular receptors for entry into the host cell. Naturally, knowing the entry receptor is essential for OVT in order to determine whether the tumor would be sensitive to a specific Ad type. Cancer cells are characterized by their ability to dysregulate the expression of many proteins and, as such, the expression of Ad receptors may differ substantially from normal cells and can sometimes vary per tumor type or disease stage. This review describes the different Ad receptors and their expression in different tumor types and disease stages, as well as following standard anticancer treatments. Moreover, the utility of Ad receptor binding will be discussed, including the implications for the selection of Ads to be used in OVT.

## 2. Adenovirus Receptor Expression in Normal Cells and Cancer Cells

Oncolytic Ads need to enter and replicate in cells in order to lyse cells. Adenoviral cell entry is a two-step process, which starts with binding to the primary receptor to ensure attachment. Binding is facilitated by the extruded fiber protein [7]. Thereupon the virus connects with integrins to initiate endocytosis and subsequent entry into the host cell. The integrins are recognized by the viral penton-base protein with the Arginine-Glycine-Aspartate (RGD) loop. The primary receptor can differ per Ad species and type (Table 1). Some species are capable of binding multiple receptors, which may enhance their infectivity from an evolutionary perspective.

As Ads use a broad range of receptors to infect cells, they do not have an intrinsic tropism to specifically infect tumors cells. Therefore, HAdVs can be modified to specifically target tumor cells. One way to restrict replication in tumor cells is by deletion of (part of) the E1A gene from the Ad genome [5]. E1A inhibits the retinoblastoma protein in cells, thereby ensuring cell cycle progression and thus replication of the Ad genome. In tumor cells, the retinoblastoma protein is often mutated, rendering this E1A function irrelevant for replication in tumor cells, but not in normal cells. As a result, the virus is able to replicate in tumor cells without a functional retinoblastoma protein. One example is the delta24-RGD Ad mutant, which has shown efficacy in several glioma mouse models and has now entered clinical trials [20]. An equivalent approach includes the deletion of E1B-55K, which seems to rely on several factors including p53 deficiency, late viral mRNA export, and/or cyclin E dysregulation [21]. Another way to induce tumor selectivity is the use of cancer-specific promotors which regulate E1A expression. All these modifications do not affect transductional targeting of cancerous cells but rather restrict their replication to these cells. A straightforward approach to improve selective tumor targeting is the use of Ad types of which the receptors are overexpressed in a tumor, as protein expression is often dysregulated in cancer. For this mechanism, it is necessary to know the expression of the oncolytic Ad receptors on tumor cells. The receptors might be downregulated, rendering oncolytic Ad therapy ineffective in this case, or upregulated, making the cancer cells a better target for oncolytic Ad therapy.

### 2.1. Coxsackie and Adenovirus Receptor

The most studied Ad receptor is the coxsackie and adenovirus receptor (CAR), and as the name states it is a common receptor for many Ads and the coxsackie B viruses [22]. More recently, CAR has also been proposed as a receptor for enteric caliciviruses [23]. Most types, including HAdV-A12, HAdV-C2, HAdV-C5, HAdV-D9, HAdV-E4, and HAdV-F41, bind to CAR [24]. The HAdV types that use CAR for attachment, bind to the receptor via the fiber protein head [25]. CAR is a 46 kDa transmembrane protein, member of the immunoglobin superfamily [7], and is part of tight junctions (TJs) in epithelial cells where it contributes to cell–cell contacts [26]. Interestingly, this localization makes it difficult for oncolytic Ads to reach the receptor and infect cells. The protein expression of CAR differs per tissue. For example, CAR expression is high in the liver, stomach, and gall bladder but is rarely detected in the thymus [27]. High expression of CAR in the liver could possibly explain the natural tropism of HAdVs, as systemically administered HAdVs sequester and transduce the liver. However, mutations in the fiber which ablate its CAR-binding abilities were shown to not affect infection of the liver [5,28]. Therefore, other factors might be at play, like the length of the fiber protein shaft [25] or binding to other receptors. In addition, the liver cells can be entered via adaptor molecules binding to the viral particle, such as factor X (see below). Reports regarding CAR protein expression are not always in agreement. For example, no staining was observed in the brain in the study of Reeh et al., whereas other reports do show CAR positive cells in some areas of the brain [29]. This variability in staining could be attributed to different staining antibodies, or due to the selected parts of the brain in the Reeh et al. study where no staining was observed. It might be coincidence that there is no staining, since there were only five samples of normal brain tissue selected in the Reeh et al. study, and Persson et al. also show no staining of the majority of the brain. Still, the overall CAR expression in the brain is low or absent, rendering the brain a poor target for CAR-binding oncolytic Ads, unless CAR is upregulated during brain tumor formation.

Most oncolytic adenoviral vectors are still based on HAdV-C5, hence knowing the expression of CAR on cancer cells is essential for successful therapy. CAR expression can be determined in several ways, for example via immunohistochemistry staining of tumor biopsies. Reeh et al. determined the protein expression of CAR in 100 malignancies as well as in healthy control tissues. They found that CAR is upregulated in a diverse range of tumor types, from lung cancer to ovarian and cervical cancer. Most of the findings about upregulated CAR expression in the Reeh et al. study were in agreement with previous reports. In addition, there were some new findings, like the upregulation of CAR in basal cell carcinoma, thyroid adenoma, and laryngeal cancer. Upregulation of CAR in these tumors makes them interesting targets for oncolytic Ad therapy with a CAR-binding virus. However, not all findings by Reeh et al. are consistent with previous reports. While upregulation of CAR in noninvasive bladder cancer was found by Reeh et al., another study described downregulation of CAR in bladder cancer [30]. The difference might be explained by the low number of samples of healthy bladder tissue used by Reeh et al., which all show low CAR protein expression, whereas the Sachs et al. healthy samples show high CAR expression. In addition, Sachs et al. state that the downregulation of CAR in bladder cancer is dependent on the cancer stage and CAR expression correlated with the invasiveness of the tumor.

Besides CAR protein upregulation, CAR can also be downregulated in some tumor types. CAR expression was downregulated in kidney, prostate, parotid gland [27], and colorectal cancers [31]. Tumors in which CAR is downregulated pose poor targets for CAR-binding oncolytic Ads. Loss of CAR expression is associated with poor prognosis and promotes metastasis due to decreased cell adhesion. CAR is therefore thought to be a tumor suppressor, although there are some contradictive results regarding this matter. In colon cancer, downregulation of CAR was associated with a higher degree of metastasis. Nevertheless, high CAR expression on the cell membrane was also necessary to establish metastasized tumors at distant sites [32]. Similar to bladder cancer, Stecker et al. suggest that CAR expression in colon cancer could be stage dependent and therefore it could be important to determine protein expression before the start of oncolytic therapy.

Downregulation of CAR in vitro is associated with oxygen shortage, which often occurs in solid tumors [33]. Hypoxia activates a transcription factor, hypoxia inducible factor-1α (HIF-1α), which indirectly inhibited mRNA synthesis of CAR in gastric, colon, and prostate cancer cell lines. Since hypoxia in tumors is generally associated with a poor prognosis, HIF inhibitors have been developed [34]. These inhibitors could be combined with a CAR-targeting oncolytic Ad to induce enhanced infection and killing of tumor cells. Many other pathways are implicated in the downregulation of CAR in tumor cells: the RAS-MEK pathway inhibits CAR expression and TGF-β signaling induces epithelial to mesenchymal transition (EMT), and thereby lowers CAR expression [35]. EMT is associated with more progressive cancers, thus CAR downregulation in this situation suggests that progressive cancers would be poorly infected by CAR-binding Ads. ZEB1 mediated inhibition of CAR transcription is essential in TGF-β regulation of CAR, although no direct link between TGF-β and ZEB1 has been found. ZEB1 is a repressor of transcription and binds to the promoter of the CAR gene, thereby inhibiting CAR transcription. In contrast, Ad infection was shown to downregulate the TGF-β1 type II receptor, thereby rendering cells less sensitive to TGF-β signaling [36]. Downregulation of TGF-β receptors would thereby prevent TGF-β induced EMT and thus CAR downregulation. However, CAR downregulation in progressive tumors that have undergone EMT would occur before the start of oncolytic Ad therapy, thereby still rendering these tumors non-permissive to therapy, even though further downregulation of CAR might be halted by TGF-β receptor downregulation during therapy. Nevertheless, the TGF-β pathway might still pose a target pathway to block CAR downregulation for early-stage tumors, who have not yet undergone EMT and still express CAR on their cell membrane. Whether the combination of TGF-β pathway inhibition and adenovirus-mediated downregulation of the TGF-β1 receptor II can indeed result in additive or synergistic effects remains to be determined. In conclusion, this and other pathways involved in CAR protein expression might still pose interesting ways of blocking CAR downregulation in tumor cells and rendering the cells susceptible to oncolytic Ad therapy with a CAR-binding virus.

In contrast to all the other Ad species, the B species does not bind to CAR, but to other receptors (Table 1). Originally, the human species B Ads are subdivided into two groups based on their genome similarity and serum neutralization. Alternatively, three distinct subgroups are formed when the species B viruses are divided based on receptor binding. The B:1 group binds primarily to CD46 (HAdV-B16, HAdV-B21, HAdV-B35, and HAdV-B50), B:2 Ads bind preferentially to desmoglein-2 (DSG-2) (HAdV-B3, HAdV-B7, and HAdV-B14), and B:3 Ads prefer to bind to CD46, however they also bind DSG-2 (HAdV-B11) [37].

### 2.2. CD46

Most species B Ad types bind to CD46 for attachment [38]. HAdV types B11, B14, B16, B21, B35, and B50 all interact with CD46, but HAdV-B3 does not use CD46 as a receptor. Moreover, some viruses from other species can also bind to CD46, like HAdV-D17 [39] and HAdV-D37 [40]. CD46 is part of the regulators of complement activation and negatively regulates the complement system [41]. In order to inhibit the complement system, CD46 cleaves complement factors C3b and C4b which are deposited on human cells. Thereby, the cells are protected from complement inflicted damage. CD46 is a transmembrane protein and mainly interacts with C3b and C4b molecules on the same cell, so there is no cleavage of C3b and C4b on other neighboring cells. Additionally, CD46 has been shown to play a role in a broad range of other processes, including its function as a receptor for the measles virus and binding of spermatozoa to oocytes. All cells with a nucleus express CD46 and therefore the tropism of HAdV from the B species is broader than that of Ad viruses using CAR as a receptor [41]. The presence of CD46 on almost all cells could be an advantage for oncolytic Ad therapy, as any cell in theory could be infected. However, CD46 has different isoforms due to alternative splicing. This might explain the specific tropism for certain regions that some virus types show. For example, HAdV-D37 specifically causes eye infections and binds preferentially to CD46 isoform C expressed in the eye [40]. Therefore, when using a CD46-binding HAdV for oncolytic therapy, it should be investigated to which isoform of CD46 the virion binds to preferentially, and whether this isoform is expressed in the tumor of interest.

CD46 expression is low or moderate in most normal tissues, but upregulated in many types of cancer [42,43]. One reason for the upregulation might be that CD46 can protect cancer cells from being killed by the complement system [44]. Upregulation of CD46 for instance, is detected in primary and metastatic prostate cancer [43], as well as in bladder cancer [42]. In the latter, CD46 expression negatively correlated with the stage of the cancer. Given that the overall expression of CD46 in normal tissue is low, CD46 would be a suitable target for oncolytic Ads binding to CD46, primarily the B species HAdVs. In cancers originating from epithelium, however, CD46 is located in cellular junctions which are hard to reach for adenoviral vectors [8]. Additionally, cells downregulate CD46 upon binding with oncolytic HAdVs. This renders CD46-targeting adenoviral vectors less preferable for long term use. In spite of these disadvantages, targeting CD46 instead of CAR with a chimeric HAdV-C5 which contained the fiber of HAdV-B35 enhanced killing of CD46 expressing colon cancer cells [45] and bladder cancer cells [42], both in vitro and in in vivo mouse models.

Knowledge on the pathways involved in CD46 upregulation could reveal new targets for combination therapy with oncolytic Ads. CD46 upregulation in cancer can partly be attributed to the activation of signal transducers and activators of transcription (STAT) 3 [46]. STAT3 binds to the promoter of CD46 and thereby induces the transcription of CD46. Knockdown of STAT3 was shown to inhibit CD46 mRNA synthesis and CD46 protein expression. STAT3 is often constitutively active in cancer cells, explaining why CD46 expression is often upregulated. In addition, p53 was shown to negatively regulate CD46 expression in myeloma cells [47]. Since p53 is one of the most frequently deleted or mutated genes in cancer, this might attribute to the upregulation of CD46 in cancer and thus making CD46 an interesting target for oncolytic Ad therapy. As a result of the active upregulation of CD46 to protect against complement induced tumor cell killing and the passive upregulation via mutations in factors such as STAT3 and p53, tumor cells form a good target for CD46-binding oncolytic Ads.

### 2.3. Desmoglein-2

DSG-2 is a transmembrane glycoprotein and a member of the cadherin superfamily. Like CAR, DSG-2 is important for cell–cell interactions and localizes to specified junctions in epithelial cells [48]. These junctions are called desmosomes and are also found in myocardium, the spleen, and lymph nodes. Like in the case of CAR and CD46, DSG-2 localization in desmosomes complicates entry into the target cell (Figure 1). In addition to expression in desmosomes, DSG-2 mRNA is detectable in the colon, bladder, prostate, liver, kidney, and stomach. Binding of Ads to DSG-2 initiates two events. Firstly, viral attachment leads to infection of the cell. Secondly, binding to DSG-2 on epithelial cells induces a process like EMT [37]. The switch to cells with a more mesenchymal phenotype results in better accessibility to Ad receptors located in narrow spots, like CAR in the TJs and DSG-2 in desmosomes, and thus enhances infectivity. Unlike Ads that bind to CAR or CD46, Ad binding to DSG-2 requires both the fiber and the penton base protein.

Similar to CD46, DSG-2 is often upregulated in cancer [56]. Upregulation of DSG-2 might seem counterintuitive in cancer, as tumors need to break cell–cell contacts in order to metastasize, but signaling via DSG-2 might promote proliferation and migration of tumor cells [57]. Additionally, DSG-2 was shown to be involved in tumor vascular formation in melanoma [58]. DSG-2 is upregulated in malignant skin cancers [56], a subset of lung cancers [59,60], colon adenocarcinomas [61], primary liver cancer [62], and cervical cancer [57]. In hepatic cancer, the upregulation of DGS-2 is positively correlated with the tumor stage and size [62]. DSG-2 expression was also associated with the tumor size in non-small cell lung cancer [59], and lung adenocarcinoma [60]. Tumor types with upregulated DSG-2 would be prime subjects for oncolytic therapy with species B Ads binding to DSG-2, like HAdV-B3 and HAdV-B7. In contrast, DSG-2 downregulation is found in primary prostate cancer, although high DGS-2 expression is seen in metastatic prostate cells lines [63]. These findings are in line with the dogma that tumor cells disrupt cell–cell interactions in order to spread through the body, but re-express adhesion molecules in order to establish secondary tumors. Additionally, downregulation of DSG-2 has been observed in gastric cancer [64], malignant ovarian tumors [65], and pancreatic cancer [66]. Downregulation of DSG-2 in pancreatic cancer decreased cell–cell contacts and increased cell invasion, promoting metastasis [67]. Not surprisingly, downregulation of DSG-2 is associated with poor clinical outcome [58,65]. This decrease in DSG-2 expression might be mediated by a protease, kallikrein 7, which is upregulated in pancreatic cancer [66]. It was shown that DSG-2 is a substrate for kallikrein 7 and upregulation of kallikrein 7 enhanced secretion of DSG-2 from cells, decreasing cell adhesion.

Since DSG-2 is mainly upregulated in tumors, the use of species B Ads binding to DSG-2 presents a promising approach in oncolytic therapy. Another advantage of DSG-2-binding Ads is the opening of intercellular spaces upon virus binding to DSG-2 [49]. Binding of HAdV-B3 to DSG-2 trigged intracellular pathways leading to the activation of a metalloprotease, ADAM17, which cleaved the extracellular part of DSG-2. Shedding of DSG-2 breaks cell–cell contacts and promotes viral spread, but might also promote easier spread of tumor cells and thereby metastasis. Taking into consideration that some tumors downregulate DSG-2 and thus form a poor target for DSG-2 binding Ads, timing is crucial in the process of DSG-2 shedding. DSG-2 shedding upon virus binding occurs on cells that are already infected, so shedding does not inhibit infection, but does open up intercellular spaces. On the contrary, overall downregulation of DSG-2 in tumor cells would prevent initial infection and render oncolytic Ad therapy inefficacious. Mechanisms for the upregulation of DSG-2 have not been elucidated, but understanding this process could be helpful in finding suitable drug targets to improve efficacy of DSG-2-binding Ads.

### 2.4. CD80 and CD86

While most human species B Ads have a preference for either CD46 or DSG-2, all species B viruses can bind to CD80 and CD86 [68]. CD80 and CD86 are co-stimulatory molecules on antigen presenting cells (APC) and participate in T cell activation [50]. Induced expression of CD80 and CD86 was shown to enhance infection of species B Ads, but not HAdV-C5 [68]. Unfortunately, few studies focus on CD80 and CD86 as receptors for HAdVs and therefore little is known about the in vivo importance of these receptors.

CD80 and CD86 are normally only expressed on APCs, but some tumor cells are able to express these co-stimulatory molecules as well [51]. Glioblastoma cells showed high expression of CD80/CD86, whereas CAR expression was low in these cells. Using a HAdV-C5 chimera vector with the fiber knob of HAdV-B3 that bound to CD80 or CD86 instead of CAR, enhanced infection of these glioma cells. In a colon cancer cell line, oxidative stress and reactive oxygen species were able to increase the expression of CD80, however elevated CD80 levels were mainly found in preneoplastic colon mucosa biopsies [52]. In contrast to many Ad receptors, CD80 and CD86 do not localize to places that are hard to reach, like CAR, CD46, and DSG-2 in between cells (Figure 1). This localization would make them an easier receptor to target with an oncolytic Ad. However, considering the expression of CD80 and CD86 on APCs, using these receptors might induce an untimely immune reaction against the HAdV, preventing infection of tumor cells in the TME. For oncolytic therapy, the immune reaction should only start once the tumor cells are properly infected. In contrast, targeting of APCs like dendritic cells is favored when using adenoviral vectors for vaccination strategies [69]. In this case using a CD80/CD86 binding Ad might enhance the efficacy of the vaccine, since an immune response against the antigen is not time-sensitive [70].

### 2.5. Sialic Acid

A receptor for several members of the human D species Ads are sialic acid (SA) residues. SA molecules consist of nine linked acidic sugars and are often attached to the end of glycans, glycoproteins, or glycolipids as a post-translational modification [53]. Synthesis of SA starts in the cytosol, from where the pre-SA is transferred to the nucleus. A cytidine-monophosphate (CMP) residue is linked to SA in the nucleus, priming it for linkage to protein or lipids. The CMP-SA locates to the Golgi, where sialyltransferases attach SA to the receiving molecule. Glycoproteins and glycolipids containing SA are ubiquitously expressed on all cells [71], thereby rendering a broad tropism of HAdVs using SA and making it an interesting receptor for oncolytic Ad therapy. However, as infection with SA-binding Ads commonly manifests in conjunctivitis, the tropism of these viruses seems to be more specific. So far, it is not clear how this typical tropism arises [7]. The conjunctivitis causing Ads are HAdV-D37, HAdV-D19a, and HAdV-D8, and bind to SA residues with their fiber knob [9]. The SA binding sequence is conserved in other Ads from the D species, suggesting that more HAdV types can bind to SA and use this as a receptor. Additionally, HAdV-G52, the only Ad member of the G species, binds to SA as well as to CAR [18]. Similar to HAdV-F40 and F41, HAdV-G52 has two types of fibers, a short fiber and a long fiber. The long fiber binds to CAR in a high affinity mode, whereas the short fiber binds to SA with lower affinity interactions. More recently HAdV-G52 was shown to bind to poly-SA of at least three SA residues [19].

Hypersialylation often occurs in cancer cells due to multiple mechanisms (as reviewed in [72]). In short, there are three mechanisms attributing to hypersialylation in cancer. Sialyltransferases transfer SA modules in the Golgi apparatus to glycans or proteins. These transferases can be overexpressed in tumor cells, increasing the amount of sialylated glycans and proteins. Additionally, the amount of substrate, CMP-SA, can be elevated in cancer cells because of changes in the metabolic influx. Lastly, the expression of neuraminidases, enzymes that cleave off the SA module, can be altered in cancer cells. Although SA is often upregulated in many cancer types, SA residues are also present on all normal cells, rendering specific tumor targeting with SA binding oncolytic Ads difficult.

### 2.6. Integrins

As stated above, Ad entry is a two-step process in which viral attachment with the primary receptor is followed by binding to integrins and internalization of the virus. Integrins are important in a myriad of cellular functions such as cell adhesion, migration, growth, and differentiation [54], and are expressed throughout the body. In addition, they function as receptors for multiple viruses, like the coxsackievirus and adeno-associated virus. Integrins form heterodimers of different α and β subunits and can be divided into four groups based on the ligands they bind. Important for HAdV cell entry is the group of the αv integrins and Ads bind to them with the RGD motif in the penton base. Interestingly, HAdV-F40 and F41 do not contain the RGD motif. Instead, HAdV-F41 binds to another group of integrins, namely the laminin-binding integrins [73]. Besides functioning as a receptor for internalization, some integrins were shown to function as primary attachment receptors [73,74]. In the absence of CAR, HAdV-C5 was still able to infect cancer cell lines via high affinity binding to the αvβ5 integrin [75]. Similarly, HAdV-D26 was shown to use the αvβ3 integrin as a receptor on epithelial cells, whereas CAR and CD46 did not play a role [74]. This questions the need for a ‘primary’ receptor for attachment.

Binding of human Ads to integrins induces virus internalization in a clathrin-dependent way [54]. The exact mechanisms that govern the endocytic uptake of the virus are still unknown, but phosphatidylinositol-3-OH kinase activation is necessary in order for internalization to take place. Subsequently, phosphatidylinositol-3-OH kinase activates Rho GTPases that in turn modify the cytoskeleton to allow viral entry. Integrins may also play a role in endosomal escape into the cytosol of Ads [7], illustrating the importance of integrin-binding in oncolytic Ad therapy. Therefore, integrin expression on cancer cells is essential for oncolytic Ad therapy. The expression of integrins is often aberrant in tumor cells, but can differ per heterodimer. αvβ3 integrins, one of the Ad-binding integrins, are overexpressed in pancreatic cancer as well as in breast cancer [76]. In addition, the expression of αvβ3 correlates with disease progression in multiple types of cancer, including melanoma, prostate, ovarian, cervical, breast, and pancreatic cancer. This means that upregulation of αvβ3 can be found in advanced stages of these types of cancer. αv integrins are also expressed on endothelial cells during angiogenesis, a crucial process in the development of tumors [76]. In order to receive enough oxygen and nutrients during extensive tumor growth, new blood vessels are formed in the TME [77]. The upregulated expression of αv integrins on these blood vessels makes them a possible target to direct oncolytic Ads to the TME. Besides enhancing viral delivery to the TME, Merchan and Toro Bejarano suggest that targeting the tumor vasculature might have an enhanced antitumor effect, as well as an antiangiogenic effect, thereby reducing the ability of tumor cells to grow [77]. However, αv integrins are also expressed on normal activated endothelial cells, which are important in wound repair [76]. In this case, the normal endothelial cells could also be infected, and this could intervene with wound repair.

Integrins are especially important in epithelial cells for cell–cell adhesion and attachment to the extracellular matrix. Some integrins, αvβ3 and αvβ6, are hardly expressed on normal epithelial cells, but upregulated on tumor tissue [55]. Thus, targeting of αvβ3 and αvβ6 with an Ad would induce specific infection of tumor cells. However, HAdVs bind to the general RGD motif in αv integrins with their penton base and assuring binding to a specific integrin heterodimer would require genetic modifications of either the penton base or the fiber knob. One example of a modified Ad binding to an integrin is Ad5-3Δ-A20T [78]. This virus was engineered to specifically replicate in tumor cells, CAR-binding was ablated, and the fiber knob was modified to bind to the αvβ6 integrins, which were upregulated in pancreatic cancer cells. Thereby, Ad5-3Δ-A20T infected and killed pancreatic tumor cells in vitro and in in vivo mouse models, showing that integrin-directed retargeting of Ads to tumor cells specifically is possible.

### 2.7. Other Receptors

Besides CAR, CD46, DSG-2, SA, and integrins, other molecules can interact with HAdVs and can function as receptors. Amongst these are scavenger receptors, like the macrophage receptor with collagenous structure (MARCO) [17]. HAdV-C5 and HAdV-C2 were shown to bind to MARCO (Table 1). Another study showed that the murine MARCO homologue is an entry receptor for HAdV-C5, HAdV-C2, HAdV-B35, and HAdV-D26 [79]. This might suggest that the human MARCO could function as a receptor for multiple Ad species. MARCO is present on macrophages, and facilitates infection of macrophages and subsequent initiation of the innate immune response against the virus [17]. This interaction might play a role in the early antiviral response and is thus of importance for oncolytic adenoviral therapy.

During tumor development macrophages travel to the TME and make up most of the immune cells that infiltrate the TME. These tumor-associated macrophages (TAMs) seem to have multiple roles, either beneficial or detrimental for the tumor [80]. For instance, TAMs can aid in tumor growth and promote angiogenesis. Contradictory, TAMs have also been shown to enhance the antitumor immune response, promoting tumor cell killing. The fact that many cells residing in the TME are macrophages, and that these macrophages express MARCO [81], presents an opportunity for oncolytic Ads binding to MARCO. As said before, infection of macrophages is essential for the start of the innate immune response against Ads and could promote antitumor immunity. However, untimely activation of the immune system could still prevent effective infection of tumor cells and thereby dampen the oncolytic effect. Additionally, MARCO is not expressed on tumor cells itself, so this questions whether tumor cells would be killed effectively when using a MARCO-binding HAdV.

HAdVs also bind to certain adapter molecules. Coagulation factor X (FX) is one of these adapter molecules and is also involved in the liver tropism of HAdVs [82]. HAdV-C5 binds to FX via the hexon hypervariable region, upon which FX binds to heparan sulfate proteoglycans (HSPGs) on liver cells. Ablating the binding of FX to HAdV-C5 diminishes virus localization to the liver. FX binding is not restricted to HAdV-C5, and many more types are capable of binding to FX, albeit with different affinities. Interestingly, all the Ads that do not bind to FX are members of the D species [82]. As homing of systemic delivered HAdVs to the liver hinders oncolytic therapy efficacy, mutating HAdV to reduce FX binding could be helpful for oncolytic therapy. However, FX does not only induce liver transduction, but also has a protective role for HAdVs in the blood stream [83]. FX binds to HAdV and forms a coat that protects against serum neutralization of virus in certain human sera. Unfortunately, there is a lot of heterogeneity in the protective role of FX partly due to differences in pre-existing immunity in the population. This shows the importance of testing the pre-existing immunity before starting oncolytic virus therapy.

FX can be detected at the protein level in some types of tumors, namely in glioblastoma, colon cancer, and endometrial cancer [84,85]. However, little research has focused on FX expression in other tumor types. Additionally, FX aids in the liver sequestration of systemically administered virus, suggesting that the exploitation of HAdV binding to FX would only enhance liver infection. Therefore, using an FX-binding Ad might be beneficial for oncolytic therapy of hepatic cancers. In the case of other types of cancer, the FX-binding should preferably be diminished in order to prevent liver sequestration and off target effects in the liver. For example, by genetic modifications or using a non-FX binding HAdV type.

Another adapter molecule for some HAdVs is dipalmitoyl phosphatidylcholine (DPPC) [86]. DPPC is the main component of pulmonary surfactant, which gives strength to the alveoli in the lung. The lungs are a preferred site for certain HAdV types, however the mechanism of infection is not completely understood. What is known, is that some HAdVs can infect the alveolar epithelium via DPPC. HAdV-C2 binds to DPPC via the hexon protein and enters alveolar cells together with DPPC, without using more common HAdV receptors. Balakireva et al. showed this by saturating the common receptors like CAR and integrins with an abundance of fiber head protein and penton base [86]. This decreased infectivity immensely, but could be rescued by adding DPPC vesicles. Nevertheless, no further research has been conducted on the role of DPPC in vivo and in the use of oncolytic Ad therapy. DPPC might pose an interesting target for OVT of lung cancers. However, in non-small cell lung cancer the expression of DPPC was lower compared to normal [87].

Similar to DPPC, only a few studies have focused on HAdV binding to vascular and cellular adhesion molecule-1 (VCAM-1). VCAM-1 expression is upregulated in atherosclerotic cells and functions as a receptor for binding of leukocytes to the vascular wall [88]. VCAM-1 shows homology to CAR and was shown to mediate HAdV-C5 infection. Although, no direct binding of HAdV-C5 to VCAM-1 was shown, suggesting lower binding affinity than to CAR [88]. VCAM-1 has been shown to be expressed on some tumor cells, whereas normally it is only expressed on endothelial cells [89], however no research has exploited this expression for oncolytic Ad therapy. A final Ad binding molecule is the major histocompatibility complex-I (MHC-I) [16]. HAdV-C5 can bind to MHC-I on human cells, however no binding is observed when MHC-I is introduced in hamster ovary cells. The role of MCH-1 in in vivo infection of HAdV is still unclear and more research is needed. More importantly, MHC-I, is often downregulated in cancer cells in order to prevent detection by the immune system [90]. This renders MHC-I less preferable as a targeting receptor in OVT.

To summarize, HAdVs can bind to a series of different molecules besides the more studied primary receptors such as CAR and CD46. Most of these Ad binding factors are not thoroughly investigated and this makes their targeting in OVT uncertain. Nonetheless, FX was shown to play a role in liver sequestering of HAdVs, a major side effect of systemic administered Ad vectors. Therefore, the binding of certain Ads to FX should not be ignored in oncolytic Ad therapy.

## 3. Adenovirus Receptor Expression after Therapy

The resilience of cancer cells, created by their ability to adapt to their environment, can also lead to alterations of the tumor after treatment. Oncolytic Ads are often combined with other therapies as the use of oncolytic Ads as a monotherapy has thus far not been very beneficial for cancer patients [91]. Oncorine, the first oncolytic Ad approved for clinical use, was specifically authorized in combination with chemotherapy [92]. Although early clinical trials showed the safety of oncolytic Ads, the efficacy remained low. Therefore, it might be important to determine the Ad receptor expression after the combined treatment if it precedes the administration of oncolytic Ads, such as with chemotherapy. Different types of chemotherapy can be combined with oncolytic Ads and these combinations showed synergistic effects and increased tumor cell killing [5]. However, chemotherapy can affect Ad receptor expression [93,94]. In one study, upregulation of CAR was seen in chemo-resistant tumor cells, suggesting that these cells upregulated CAR due to chemotherapy treatment [94]. In another study it was shown that chemotherapy induced cell senescence and these senescent cells decreased the expression of CAR on the membrane [93]. In the case of upregulated CAR, monotherapy with an oncolytic Ad showed increased effectivity in killing chemo-resistant breast cancer cells [94]. The variation in up- or downregulation of CAR in reaction to chemotherapy could be attributed to different chemotherapies used in these studies. It would be worthwhile to determine whether oncolytic Ads can be combined with every chemotherapeutic agent and to investigate how it affects receptor expression.

In addition to chemotherapy, radiotherapy makes a promising partner for combination therapy with oncolytic Ads. Radiation kills cells by inducing DNA damage which ultimately leads to cell death [95]. However, cells possess intrinsic mechanisms to repair DNA damage, limiting the effect of radiotherapy. This is where oncolytic Ads can be put to use, as HAdV can interact with and inhibit the DNA repair mechanisms, thereby enhancing the effect of radiotherapy. Conversely, radiotherapy might enhance viral oncolysis of tumor cells by increasing viral uptake or replication [96]. Clinical trials with the combination of an oncolytic Ad and radiotherapy confirmed safety, although clear enhanced efficacy of the combination therapy has not yet been shown [95]. This might be due to downregulation of Ad receptors, which could be an effect of the radiation induced DNA damage. However, little is known about the effects of radiotherapy on Ad receptors. C. Liu et al. showed that radiotherapy had no effect on the expression levels of CD46 on glioblastoma cells [97]. By contrast, the results regarding the effects of radiotherapy on CAR expression are conflicting [96]. One study showed no effect on CAR expression in glioblastoma cell lines, whereas another study showed upregulation of CAR in colorectal and head and neck cancer cell lines. This difference might be tumor type related and, therefore, more research is needed on the effect of radiotherapy on the Ad receptor expression in various tumors in order to improve clinical benefit of combining both therapies.

Knowing the effect of chemotherapy or radiotherapy on Ad receptor expression is pivotal during the development of oncolytic Ads. Experimental therapies, like oncolytic Ad therapy, are only used after the standard treatment fails. The standard treatment for many cancers is still either chemotherapy or radiotherapy. Should these treatments have a negative effect on the receptor expression of certain HAdVs, this might put a stop to the development of effective oncolytic Ad vectors that target these specific receptors. Therefore, for radiotherapy or chemotherapy to enter the clinic as an efficient combination therapy with different oncolytic Ad vectors, the effects of these therapies on Ad receptor expression should be better characterized.

## 4. The Multifaceted Role of Adenovirus Receptors

### 4.1. Viral Spread

Besides being essential for virus attachment to the cell, Ad receptors have been suggested to play a role during release of new virions. However, not much is known about the exact role of the different receptors during this process. After binding to the surface receptors, internalization, and replication, Ads exit the cell in order to infect neighboring cells. Some primary receptors are located at places that are hard to reach for the virus, such as CAR, DSG-2, and CD46 on epithelial cells. Although there are little structural similarities between CAR, DSG-2, and CD46 and in their manner of binding to the HAdV fiber knob, CAR and DSG-2 have a similar function as cell adhesion proteins [98]. Binding to cellular adhesion molecules is conserved between a lot of viruses due to their abundance throughout the human body [99]. However, using cell adhesion molecules as primary attachment receptor seems counterintuitive, as these receptors are located in places that are hard to reach, like TJs (Figure 1). Therefore, it was hypothesized that these primary attachment receptors also play a role in the viral release, especially release from epithelial cells, where cell–cell interactions are abundant [100]. Ad et al. suggested that HAdVs first bind to more accessible receptors to infect cells, but that the binding to CAR and DSG-2 is necessary for viral release and further lateral spreading of the infection [49], which is essential for OVT in order to kill the majority of the tumor. The same phenomenon is observed with herpes simplex virus (HSV) [101]. Like Ads, HSV is capable of binding multiple receptors including nectin-1 and nectin-2. The nectins are cellular adhesion proteins present in epithelial cell junctions, and like the CAR part of the immunoglobulin superfamily. It was shown that HSV binding to nectin promoted cell to cell spread in cell monolayers. Spear suggests that epithelial cells are first infected by HSV binding to a more accessible receptor, and cell to cell spread is promoted via nectin binding [101]. The latter is in agreement with overproduced fiber knobs and nonreplicating viral particles during Ad infection observed by Walters et al. [100]. The excess of fiber was secreted from the cell and interfered with CAR–CAR interactions between two cells (in the case of CAR-binding Ads). Consequently, the cell–cell interactions were disrupted, creating more space between epithelial cells. The newly released virions could better reach the apical side and promote infection to other areas. Although the fiber molecules were released in a nonlytic way, the process of disrupting cell–cell interactions to promote infection might be of great importance in oncolytic virus therapy. As CAR is also expressed on endothelial cells, disrupting endothelial CAR–CAR interactions is suggested to promote spread of the virus to the blood stream [100], which would enhance oncolytic Ad therapy by spread of the virus to possible metastases. Controversially, breaking cell–cell contacts between endothelial cells could also have negative effects in the case of tumor metastasizing. Disrupted blood cell lining might make it easier for tumor cells to reach the blood stream and metastasize.

Viral interactions with DSG-2 can be exploited as well to promote lateral spread during adenoviral infections. During replication of Ads binding to DSG-2 dodecahedral particles are produced [37]. These particles contain the penton base and fiber proteins, but not the double-stranded DNA and are therefore not infectious. Wang et al. showed that release of the dodecahedral particles interfered with DSG-2 interactions between cells, opening up the intercellular space. This could result in easier access to receptors in order to promote lateral spread. HAdV-B3 is one of the DSG-2 binding viruses and this interaction led to the development of a therapeutic molecule with Ad3 fiber knobs to open up intercellular spaces in epithelial cell layers [8]. This therapeutic molecule, junction opener-1, could be combined with oncolytic Ads that target receptors located in between cells, to promote easier access to these receptors and thereby enhance the infectivity. The hypothesis that Ads use intercellular receptors to ensure spreading to other cells, would require binding of Ads to another, more accessible receptor. However, as far as we know, not all HAdV types are capable of binding to multiple primary receptors. For example, HAdV-E4 and HAdV-F40 have been shown to bind to CAR only (Table 1). A possible explanation would be that these Ad types infect cells via apically localized CAR [102]. CAR has two isoforms due to alternative splicing, CAR^ex7^ and CAR^ex8^. Both isoforms localize to cellular junctions in nonpolarized cells, but in polarized cells CAR^ex8^ localizes to the apical side as opposed to CAR^ex7^, which remains in cellular junctions. It was shown that expression of CAR^ex8^ promotes adenoviral infection from the apical side, although most tested cell lines showed lower CAR^ex8^ expression compared to CAR^ex7^. The presence of apically located CAR^ex8^ could improve oncolytic Ad therapy and exploiting mechanisms to induce this apical expression might prove promising for combination therapy strategies.

Another explanation for adenoviral binding to receptors located in intercellular junctions might be that we simply have not yet characterized all the receptors used by the different Ad types. This is true, especially, for the more recently developed non-human primate adenoviral vectors [15]. HAdV could bind to a more accessible entry receptor and use the cellular adhesion receptors for lateral spread. Additional receptors for certain HAdV types are still being discovered [19,79], suggesting that we may have only scraped the surface on the vast array of Ad receptors. This might indicate that more HAdV types are capable of binding to multiple receptors in order infect cells and convey further cell to cell spread. If so, these HAdV could be valuable for solid tumors especially. So far, OVT has shown moderate efficacy in the treatment of solid tumors, presumably due to limited spread within the tumor. Several attempts have been made to increase intratumor spread, by i.e., increasing syncytium formation [103] or in combination with immunotherapy [104]. However, the use of alternative receptors might provide a simpler solution. Members of the species D Ads already demonstrated improved viral spread compared to HAdV-C5, which was attributed to their expanded tropism [105]. As such, insight into these additional receptors and their expression on tumor cells could be paramount for successful oncolytic Ad therapy. Taken together, Ad receptors located in intercellular junctions might play another role besides enabling viral entry into the host cell. The additional function of some Ad receptors, like CAR and DSG-2, to facilitate lateral spread of the virus could possibly play a more dominant role compared to their function as primary entry receptor, within the context of OVT. For solid tumor especially, Ads binding to these receptors might be superior as oncolytic agents.

### 4.2. Broad Tropism

As described in this review, HAdVs generally bind to a multitude of receptors. This raises the question what the added benefit to a virus is of targeting multiple receptors. Binding to multiple different receptors might give the virus an advantage from an evolutionary perspective, as the host will most likely not be able to prevent binding of a virion to all the different receptors at once. This wide range of receptors might have arisen from an evolutionary battle between host and virus to escape or facilitate infection, respectively. To illustrate, host receptors can sometimes mutate in such a way that the primary function is maintained, while virus binding is inhibited. As a counter-reaction, viruses mutate their cellular binding proteins in order to target different receptors. In addition, due the high pre-existing immunity against different Ad types, there is a constant immunogenic pressure for the virus to mutate and prevent neutralization, similar to the antigenic drift of influenza A virus [106]. Spontaneous mutations that attribute to immune escape might induce a change of receptor preference, thereby contributing to the diverse group of Ad receptors.

Binding to multiple receptors ensures a broad tropism inside the human body, as many tissues can be infected by HAdvs [7,8]. This ensures replication of the virus, in order to produce progeny and establish further spread. This broad tropism is beneficial for oncolytic Ad therapy, since many tissues and tumors derived from these tissues express the Ad receptors and can therefore be targeted by oncolytic therapy. However, it should be noted that the mere presence of the receptor at many sites does not lead to infection of all these tissues, per se. Different receptor isoforms might ensure a more specific tissue tropism, as is the case for CD46. Additionally, intracellular conditions need to allow for establishment of a viral infection. As a solution, the binding of one HAdV type to multiple receptors, like HAdV-G52 which binds to CAR and poly SA residues, could be exploited for oncolytic Ad therapy. Therapy efficacy might be enhanced by using such a vector, as multiple cell types or tissues could be infected with the use of only one HAdV vector.

In conclusion, the binding of HAdVs to a multitude of receptors ensures infection and replication in the host and additionally, renders a broad tropism which OVT can use to its benefit. Further characterization of the receptors that are used by the different Ad types currently being developed as oncolytic agents might provide some new insights into the variations in efficacy observed between different adenoviral vectors and tumor types. The ability of some HAdV to bind to multiple receptors, of which some ensure further cell–cell spread, seems promising for their use as oncolytic agents. However, whether this also results in improved tumor killing and/or the induction of a more potent antitumor immune response remains to be elucidated.

## 5. Future Directions

Oncolytic Ad therapy poses a new interesting treatment option for cancer patients. The different HAdV types bind to a diverse array of receptors and provides them with a broad tropism, which could be useful in OVT. However, the HAdV receptors are differentially regulated in tumors compared to normal tissues. Knowing the receptor expression of Ads on different tumor types and stages can help determine which tumor types may be susceptible to oncolytic Ad therapy. This was recently illustrated by efficacy of a chimeric oncolytic Ad5/3 vector in ovarian cancer, which uses DSG-2 as its dominant entry receptor [107]. Treatment with this specific Ad in a phase I clinical trial was suggested to considerably increase overall survival. The vector, named ONCOS-102, demonstrated effective oncolysis of several ovarian cancer cell lines, while in the absence of DSG-2, despite the presence of CD46 and CAR, no oncolysis was observed. In the future, it would therefore be best to screen tumor tissue for the presence of Ad receptors. Moreover, their additional role in cell to cell spread of the virus might make their presence a requisite for OVT to be effective. Screening of every patient before the start of therapy would make oncolytic Ad therapy a more personalized medicine instead of an off-the-shelf therapy available to every cancer patient.

Besides screening tumors, knowing the mechanisms behind either the up- or downregulation of receptors in tumor cells might elucidate new targets for combination therapy. This is very important, as oncolytic Ad therapy so far has not proved successful as a monotherapy. Unfortunately, the mechanisms behind the upregulation of CAR have not been studied widely, even though this could uncover new targets for combination therapy. Especially since some tumors downregulate CAR and most adenoviral vectors are still based on the CAR-binding HAdV-C5. Instead, research now aims to use other types of Ads, like non-human primate Ads [15] or genetically modified vectors that bind to an upregulated receptor on cancer cells [5]. However, regulations associated with genetically modified viruses complicate the approval for clinical use [108]. Moreover, genetically modified Ads, like the chimeric ONCOS-102 and Ad5/35 targeting CD46 instead of CAR, are often based on the HAdV-C5 vector and pre-existing immunity against HAdV-C5 results in reduced effectiveness of oncolytic Ad therapy [13].

Species B Ads pose interesting candidates for oncolytic therapy as their receptors, CD46 and DSG-2, are often upregulated in cancer. CD46 protein expression was reported to be low in normal tissues, which would limit the infection of normal cells during therapy and thereby reduce side effects. Moreover, CD46 is localized all over the cell membrane, in contrast to CAR (https://www.proteinatlas.org/) and binding to DSG-2 could augment lateral viral spread. HAdV-B11 might pose a good candidate for future research, as it binds to both CD46 and DSG-2, and the seroprevalence is low in the human population. However, liver sequestering via binding to FX could still form an issue. HAdV-D17, in contrast to HAdV-B11, binds to CD46 and not to FX. This suggests that HAdV-D17 might efficiently infect tumor cells, while circumventing off-target effects in the liver. Unfortunately, pre-existing immunity against HAdV-D17 is still quite high, around 30% [10]. In addition, little is known about the oncolytic potential of these Ads, hence it remains to be determined whether their oncolytic potential is comparable to that of HAdV-C5. Collectively, this highlights the complexity of finding a less seroprevalent HAdV that targets upregulated receptors in cancer patients.

The binding of HAdVs to multiple receptors ensures lateral spread, viral replication, and broad tissue tropism, which can be exploited by oncolytic Ad therapy. The generation of a broad panel of Ad vectors that bind to different receptors poses a promising way to treat many cancer patients with a range of different tumor types. Instead of focusing on the well-studied HAdV-C5 vector, less seroprevalent, wildtype-like vectors binding to upregulated receptors in tumors might enhance oncolytic Ad therapy efficacy. From the evidence provided in this review, we believe that Ads other than HAdV-C5, that target for instance CD46 or DSG-2, would be most suitable for the targeting and killing of the majority of tumor types. Nevertheless, the future of oncolytic Ad therapy lies in the combination with other therapies. Either new therapies targeting the up- or downregulation of Ad receptors or more conventional therapies like chemo- and radiotherapy. Irrespective of the combinational therapy, their effects on the expression of Ad receptors should be considered, to optimize combination therapies using oncolytic Ads and to increase their efficacy as anticancer treatments.

## Figures and Tables

**Figure 1 ijms-21-06828-f001:**
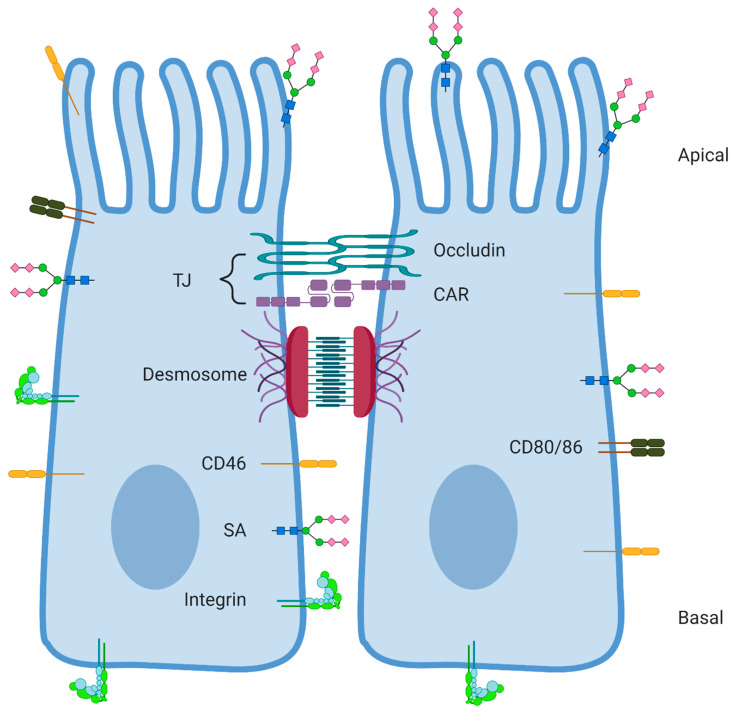
Localization of the main discussed receptors for human adenoviruses (HAdVs) on epithelial cells. Epithelial cells are polarized and contain an apical and basal side [26]. Tight junctions (TJs) and desmosomes keep the cells close together. Coxsackie and adenovirus receptor (CAR) and some other proteins like occludin make up the TJ. Desmoglein-2 (DSG-2) is part of the desmosome structure, localized below the TJ [49]. CD46 protein staining shows localization all over the cell membrane (https://www.proteinatlas.org/), but on epithelial cancer cells CD46 is found localized in between cells [8]. CD80 and CD86 are normally not expressed on epithelial cells, but on antigen presenting cells (APCs) such as dendritic cells [50]. However, some tumor cells have been shown to upregulate CD80 or CD86 [51,52]. Sialic acid (SA) residues are often the final residues on glycans and glycoproteins distributed all over the cell membrane [53]. Integrins are located all over the cell membrane in order to attach to the basement membrane, ensure cell–cell contacts, and adhere to the extracellular matrix [54,55]. Created with BioRender.com (Accessed 14 September 2020).

**Table 1 ijms-21-06828-t001:** Overview of the known primary receptors of adenovirus species and types discussed in this review.

Species	Types	Receptors
A	12, 31	CAR ^1^
B:1	3, 7, 16, 21, 50	CD46, CD80, CD86, DSG-2
B:2	11, 14, 35	CD46, CD80, CD86, DSG-2
C	1, 2, 5	CAR, αvβ5 integrin, HSPG, VCAM-1, MHC-Iα2 [16], MARCO [17]
D	8, 9, 17, 19, 37, 48	CAR, SA, CD46
E	4	CAR
F	40, 41 ^2^	CAR
G	52	CAR [18], poly-SA [19]

^1^ Abbreviations: CAR: coxsackie and adenovirus receptor. DSG-2: desmoglein-2. HSPG: heparan sulphate proteoglycan. VCAM-1: vascular cell adhesion molecule 1. MHC1α2: major histocompatibility complex-1 α2 domain. MARCO: macrophage receptor with collagenous structure. SA: sialic acid. Table composed from [7,8] unless stated otherwise. ^2^ Only the long fiber proteins of HAdV-F40 and HAdV-F41 bind to CAR.

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
