# Peer review of "Adenovirus Receptor Expression in Cancer and Its Multifaceted Role in Oncolytic Adenovirus Therapy"

_ijms, 2020, doi:10.3390/ijms21186828_

Round 1

Reviewer 1 Report

Hensen, L., Hoeben R. and Bots, S.; Review; Adenovirus receptor expression in cancer and its multifaced role in oncolytic adenovirus therapy. (MS No. IJMS-930406)

There are three indispensable steps for oncolytic virus to kill targeted cancer cells; infection, replication, and destruction by producing themselves.  The authors reviewed adenoviral receptors which is one of the most important steps for oncolytic virotherapy. This review already covered broad of receptors and balanced.  However, there are some points that require clarification before the manuscript can be considered suitable for publication.

  1. The authors wrote the details of receptor of adenovirus. There should be more graphic scheme for understanding.

  1. Section 1; The authors emphasized the relationship between CAR expression and TGFb signaling. Adenoviral infection upregulates an expression of TGFb receptor.  What is conclusion in section 1? The more progressive cancer cells become, the more efficiently oncolytic adenovirus infects?

Author Response

First, we would like to thank the reviewers for their constructive feedback.

Response to reviewer 1

There are three indispensable steps for oncolytic virus to kill targeted cancer cells; infection, replication, and destruction by producing themselves.  The authors reviewed adenoviral receptors which is one of the most important steps for oncolytic virotherapy. This review already covered broad of receptors and balanced.  However, there are some points that require clarification before the manuscript can be considered suitable for publication.

  1. The authors wrote the details of receptor of adenovirus. There should be more graphic scheme for understanding.

We fully agree that Figure 1 should show the additional receptors that are discussed in the manuscript. There are additional receptors like sialic acids, CD80/C86, and integrins and these form potential targets for specific tumor types. For example, CD80/CD86 is specifically upregulated in glioblastoma (as described in line 355) and might form a better alternative compared to CAR, CD46, and DSG-2. Therefore we have amended Figure 1 to include all receptors that have been addressed individually in the manuscript, which include CAR, CD46, DSG-2, sialic acids, CD80/CD86, and integrins. We have not included the receptors which are discussed in section 2.6 ‘Other Receptors’ which include MARCO, coagulation factor X, DPPC, V-CAM, and MHC-I as they either include adaptor molecules which are not expressed on the cells surface, have only been shown to bind to specific adenovirus types, or have not been thoroughly investigated which makes their targeting in OVT uncertain.

  1. Section 1; The authors emphasized the relationship between CAR expression and TGFb signaling. Adenoviral infection upregulates an expression of TGFb receptor. What is conclusion in section 1? The more progressive cancer cells become, the more efficiently oncolytic adenovirus infects?

We have included a brief discussion on the role of the TGFb receptor during adenoviral infection and its influence on CAR receptor expression (line 212-226). Adenoviruses have indeed been shown to downregulate the TGF-β1 receptor II during infection, thereby rendering the cells less sensitive to TGF-β and inhibiting downregulation of CAR. However, progressive tumors would have undergone EMT and CAR downregulation before the onset of therapy with an oncolytic adenovirus (line 207). These tumor would then still pose poor targets for CAR-binding oncolytic adenoviruses. The TGF-β pathway might still pose a target pathway to block CAR downregulation for early stage tumors, who have not yet undergone EMT and still express higher levels of CAR on their cell membrane. However, whether the combination of TGF-β pathway inhibition and adenovirus-mediated downregulation of the TGF-β1 receptor II can indeed result in additive or synergistic effects remains to be determined.

Reviewer 2 Report

The authors review adenovirus receptor expression in cancer and normal tissue. Adenoviruses are the most common vectors for gene therapy and can easily be modified to be oncolytic – or cancer specific. Most adenovirus vectors are based on type 5. However, the CAR receptor for Ad5 might be downregulated in some tumors. In addition, existing neutralizing antibodies might reduce the efficacy. Thus, many are switching to other adenovirus types or modifying the fiber knob to target other receptors. In addition, the modifications might overcome the problem of vector neutralization. This review summarizes the adenovirus receptor expression in different tumor types. It additionally discusses about the effects of conventional therapies to receptor expression and the rational for combining these with adenovirus treatments.

The manuscript is well structured and multifaceted. The authors describe the well-established Ad receptors and those of which limited knowledge exists. Discussion on clinical use of oncolytic adenoviruses remains shallow, but the molecular mechanisms are well argued.

Chapter 1 raises the concern of neutralizing antibodies hampering the oncolytic virus therapy efficacy. However, there are some clinical and preclinical data showing that nAbs are not hampering the OVT efficacy as much as thought (e.g. Li et al. 2017 Clin. Cancer Res., vol. 23, no. 1, pp. 239-249. and Ricca, et al. 2018, Mol. Ther. Apr 4;26(4):1008-1019 and others). Discussion could be added.

Chapter 2 presents oncolytic adenoviruses and their receptors. Partial deletion of E1A is presented as one modification making adenoviruses selective to malignant cells. In addition, there are two other approaches to make adenoviruses selective: cancer-specific promoters (such as hTERT) and E1B deletions. Discussion could be added.

Figure 1 could show the localization of all receptors discussed in the paper.

A table listing the cancer types and normal tissues where the receptors are upregulated (and downregulated) would be useful and add structure to the paper.

Chapter 3 could mention that the first (and only) oncolytic adenovirus approved for clinical use, Oncorine, was approved specifically in combination with chemotherapy.

Chapter 5 discusses future directions for oncolytic adenoviruses. Currently, there are several chimeric adenoviruses under development, which could be presented in the review. The advantages/disadvantages of using these could be brought up.

Rows 668-673 discuss about ONCOS-102. The virus is actually not HAdV-B3, but a 5/3 chimera (Ad5 with Ad3 fiber knob). Needs to be corrected.

Author Response

First, we would like to thank the reviewers for their constructive feedback.

Response to reviewer 2

The authors review adenovirus receptor expression in cancer and normal tissue. Adenoviruses are the most common vectors for gene therapy and can easily be modified to be oncolytic – or cancer specific. Most adenovirus vectors are based on type 5. However, the CAR receptor for Ad5 might be downregulated in some tumors. In addition, existing neutralizing antibodies might reduce the efficacy. Thus, many are switching to other adenovirus types or modifying the fiber knob to target other receptors. In addition, the modifications might overcome the problem of vector neutralization. This review summarizes the adenovirus receptor expression in different tumor types. It additionally discusses about the effects of conventional therapies to receptor expression and the rational for combining these with adenovirus treatments.

The manuscript is well structured and multifaceted. The authors describe the well-established Ad receptors and those of which limited knowledge exists. Discussion on clinical use of oncolytic adenoviruses remains shallow, but the molecular mechanisms are well argued.

  1. Chapter 1 raises the concern of neutralizing antibodies hampering the oncolytic virus therapy efficacy. However, there are some clinical and preclinical data showing that nAbs are not hampering the OVT efficacy as much as thought (e.g. Li et al. 2017 Clin. Cancer Res., vol. 23, no. 1, pp. 239-249. and Ricca, et al. 2018, Mol. Ther. Apr 4;26(4):1008-1019 and others). Discussion could be added.

We agree with the argument that the impact of neutralizing antibodies on the efficacy of oncolytic immunotherapy is indeed important as the effects have shown to be quite counterintuitive sometimes. For example, Berkeley et al. (2018) demonstrated for reovirus how neutralizing antibodies can lead to improved anti-tumor responses due to the monocyte-mediated hand-off of infectious reovirus to the tumor. Similar observations could be made for Coxsackievirus but not HSV, which illustrates the variability between viruses. For adenoviruses, there is no consensus yet but the presence of neutralizing antibodies is generally believed to be disadvantageous . The literature that you cited here describes the effect of neutralizing antibodies on the efficacy of OVT when administered intratumorally. We believe these support our statement that ‘nAbs mainly hinder systemic administration of oncolytic HAdVs as opposed to intratumoral injection’ (line 67). As such we have cited the article by Li et al in line 68. Furthermore, as described in the manuscript, there are several advantages of systemic administration over intratumoral injection (line 72-75). While the effect of neutralizing antibodies on systemic administration of adenoviral vectors remains to be further elucidated it could desirable to evade pre-existing immunity in order to create a more homogenous patient population. This might reduce the observed patient-to-patient variation in clinical trials, regardless of the effect of neutralizing antibodies. This has been added this to the manuscript in line 75-79.

  1. Chapter 2 presents oncolytic adenoviruses and their receptors. Partial deletion of E1A is presented as one modification making adenoviruses selective to malignant cells. In addition, there are two other approaches to make adenoviruses selective: cancer-specific promoters (such as hTERT) and E1B deletions. Discussion could be added

We have added a small discussion on the two other approaches you suggested, which include the E1B deletions and cancer-specific promotors (line 132-141). All these approaches restrict the viral replications to cancerous cells as opposed to healthy cells. Nevertheless, these modifications do not affect the transductional targeting to tumor cells, that could enhance their efficacy even further. Therefore, it remains worthwhile to select adenoviruses which have a natural preference for a receptor that is highly expressed on the tumor, in addition to the approaches mentioned here.

  1. Figure 1 could show the localization of all receptors discussed in the paper.

We fully agree that Figure 1 should show additional receptors that are discussed in the manuscript. The manuscript describes how CAR, CD46, and DSG-2 are the most prominent receptors for the different adenovirus species, and possibly the most interesting targets for oncolytic virotherapy. Nevertheless, we agree that there are additional receptors like sialic acids, CD80/C86, and integrins and might constitute plausible targets for specific tumor types. For example, CD80/CD86 is upregulated in glioblastoma (as described in line 355) and might constitute a better alternative than CAR, CD46, and DSG-2. Therefore, we have amended Figure 1 to include all receptors that have been addressed individually in the manuscript, which include CAR, CD46, DSG-2, sialic acids, CD80/CD86, and integrins. We have not included the receptors which are discussed in section 2.6 ‘Other Receptors’ which include MARCO, coagulation factor X, DPPC, V-CAM, and MHC-I as they either include adaptor molecules which are not expressed on the cells surface, have only been shown to bind to specific adenovirus types, or have not been thoroughly investigated which makes their targeting in OVT uncertain.

  1. A table listing the cancer types and normal tissues where the receptors are upregulated (and downregulated) would be useful and add structure to the paper.

We would agree that a table consisting of the expression of the receptors discussed in different tumor types could be informative but are cautious for it might lead to misinterpretation of the data presented in the manuscript. Firstly, it would be difficult to summarize the expression per tumor type or disease stage as not all tumor types are discussed for all the different receptors and some tumor types are discussed as a specific tumor type such as lung adenocarcinoma, whereas sometimes tumor types are summarized under a general term such as ‘colon cancer’ or ‘lung cancer’. Secondly, the adenovirus receptors are often up- or downregulated dependent on the tumor stage and this differs then again per tumor type (line 185, line 196, line 260, line 315, line 427). Moreover, the effects of combinations treatments like chemo- and radiation therapy can also differentially alter the expression of the receptors and so far there is no consensus on their effect on adenovirus receptor expression (line 540-543 and line 559-563) which might be caused by the difference in tumor type (line 563). Taken together, we are concerned that summarizing  the up- or downregulation of receptors on different tumor types in a table would oversimplify the matter. The manuscript indirectly highlights this variability by its proposition to screen patients before treatment for the expression of several adenovirus receptors (line 707) and to develop a panel of different adenovirus species for OVT as to be able to target a broad spectrum of tumor types (line 739).

  1. Chapter 3 could mention that the first (and only) oncolytic adenovirus approved for clinical use, Oncorine, was approved specifically in combination with chemotherapy.

Thank you for the suggestion. We have added the use of Oncorine in combination with chemotherapy in chapter three (line 533).

  1. Chapter 5 discusses future directions for oncolytic adenoviruses. Currently, there are several chimeric adenoviruses under development, which could be presented in the review. The advantages/disadvantages of using these could be brought up.

The development of chimeric adenoviruses, usually consisting of a HAdV-C5 backbone with a fiber from a species B adenovirus like HAdV-B3 or HAdV-B35, has gained much attention to resolve the (re)targeting of tumors without CAR expression. The advantage of these vectors is that they display a broader tropism due to their binding to more ubiquitously expressed cellular proteins (like CD46 or DSG-2) which makes these viruses applicable to a range of different tumor types. Their effectiveness has been illustrated in both pre-clinical (line 266) and clinical research (line 699), as mentioned in the manuscript. The main disadvantage of the use of such vectors is that, due to the HAdV-C5 backbone, any undesirable effects resulting from pre-existing neutralizing antibodies against HAdV-C5 are still of concern. We have highlighted this issue regarding these modified adenoviruses in line 719, which has been explained further in the introduction (line 67-79).

  1. Rows 668-673 discuss about ONCOS-102. The virus is actually not HAdV-B3, but a 5/3 chimera (Ad5 with Ad3 fiber knob). Needs to be corrected.

Thank you for pointing out this error. We corrected it.